# Intravital imaging-based analysis tools for vessel identification and assessment of concurrent dynamic vascular events

Naoki Honkura [1], Mark Richards[1], Bàrbara Laviña[1], Miguel Sáinz-Jaspeado[1], Christer Betsholtz[1] & Lena Claesson-Welsh [1]

The vasculature undergoes changes in diameter, permeability and blood flow in response to specific stimuli. The dynamics and interdependence of these responses in different vessels are largely unknown. Here we report a non-invasive technique to study dynamic events in different vessel categories by multi-photon microscopy and an image analysis tool, RVDM (relative velocity, direction, and morphology) allowing the identification of vessel categories by their red blood cell (RBC) parameters. Moreover, Claudin5 promoter-driven green fluorescent protein (GFP) expression is used to distinguish capillary subtypes. Intradermal injection of vascular endothelial growth factor A (VEGFA) is shown to induce leakage of circulating dextran, with vessel-type-dependent kinetics, from capillaries and venules devoid of GFP expression. VEGFA-induced leakage in capillaries coincides with vessel dilation and reduced flow velocity. Thus, intravital imaging of non-invasive stimulation combined with RVDM analysis allows for recording and quantification of very rapid events in the vasculature.

[1] Department of Immunology, Genetics and Pathology, Rudbeck Laboratory, Science for Life Laboratory, Uppsala University, Dag Hammarskjöldsv. 20, 751 85 Uppsala, Sweden. Correspondence and requests for materials should be addressed to N.H. (email: pon@hama-med.ac.jp) or to L.C-W. (email: lena.welsh@igp.uu.se)

The vasculature has essential functions in normal adult physiology and contributes to diseases by participating in inflammation and by supporting growth and spread of cancer[1,2]. While blood vessel growth and remodeling develop over days, changes in vascular size and permeability can occur in seconds. Inflammatory cytokines such as histamine and brady-kinin, as well as certain growth factors such as vascular endothelial growth factor A (VEGFA) induce rapid vascular leakage and changes in vascular caliber and blood flow velocity[3]. Vascular caliber and blood flow velocity influence static vascular sieving, allowing small molecules to continuously extravasate into the extravascular environment[4,5].

Inflammatory cytokines and VEGFA increase vessel permeability[6,7], by inducing loosening of adherens junctions (AJs), leading to the extravasation of solutes and macromolecules[7–9]. Extravasation of inflammatory cells also involves changes in AJs[10]. Postcapillary venules have been considered the main sites of leakage in most organs[11,12] but in the central nervous system (CNS), a greater abundance of endothelial tight junctions (TJs) contributes to restrictions in blood–tissue exchange imposed by the blood–brain barrier[13].

In pioneering work by Krogh[14] and subsequently, by Pappenheimer[15], Michel[16,17] and others, vascular dynamics were examined by light microscopy in isolated capillaries, perfused muscles, skin, and mesenteric vessels, revealing capillary dilatation and sieving of molecules. Locations and routes of transendothelial extravasation of macromolecules were further elucidated by Palade[18–20], Dvorak[21], McDonald[11] and others using transmission electron microscopy. This work collectively provided the background for the current study using high-resolution live imaging under atraumatic conditions to reveal properties that govern the dynamic response of specific regions of the vasculature to stimuli such as VEGFA.

Advanced imaging techniques have increased the understanding of vascular contributions to disease processes and effects of various vessel-targeting therapies[22]. Multi-photon laser scanning microscopy (MPLSM) allows highly sensitive imaging of small vessels at depths up to 1.3 mm[23,24]. Kamoun et al.[25] developed the MPLSM-based relative velocity field scanning (RVFS) methodology to follow individual, fluorescent-dye-labeled, transplanted red blood cells (RBCs) and measure blood flow in vascular vessel networks at high resolution. However, RVFS is not compatible with the concurrent visualization of different rapid processes as it requires scans at multiple angles. MPLSM has also been used to visualize the dynamics of vascular reactions of the exposed dermis of ear skin[26] and the CNS neurovascular units using headbar and cranial window implantations[27]. Since complex invasive procedures may be accompanied by inflammatory reactions with impact on the vascular response, we sought to examine vascular responses in a non-invasive manner.

Here, we describe a MPLSM method to non-invasively record rapid changes in the vasculature of the ear dermis. Time-lapse imaging is used to follow dynamic changes in vessel diameter, leakage, and RBC velocity simultaneously, before and after administration of endothelial agonists by intradermal injection through a fine glass capillary. An image analysis tool, RVDM (relative velocity, direction, and morphology), developed to monitor these rapid changes in specific types of vessels, relies on the laser scan speed and relative velocity, and direction of blood flow, which produce distinct RBC images during acquisition. Application of these technologies on a genetic mouse model where green fluorescent protein (GFP) is expressed in endothelial cells, controlled by the Claudin5 (Cldn5) promoter, reveals that capillaries and postcapillary venules, with no or low Cldn5 expression, respond to VEGFA with rapid changes in blood flow, vessel tone, and leakage. This approach provides important insights into vessel-specific dynamics in normal skin.

## Results

**Experimental setup for in vivo imaging of vascular dynamics.** To follow vascular dynamics over time, we established a setup for non-invasive intravital time-lapse imaging using MPLSM. Imaging was performed on the ventral aspect of the C57BL/6 mouse ear dermal vasculature by fixing to a glass slide (Fig. 1a). Vessels were visualized by systemic administration of fluorescent tracers via tail vein cannulation. A sub-micron glass capillary was used

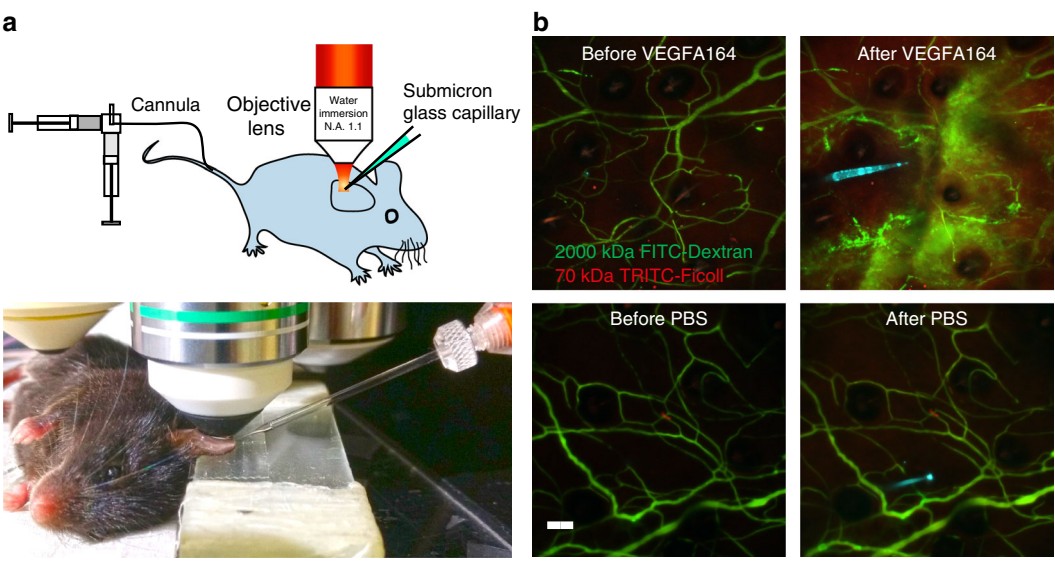

**Fig. 1** Non-invasive imaging of the ear dermis vasculature. **a** Mice were anesthetized and immobilized to fix the ear under a water-immersion objective lens for intravital imaging. Cannulation of the tail vein allowed administration of fluorescent labeled dextrans or other carriers into the circulation. A sub-micron glass capillary was used for atraumatic intradermal injection. **b** Examples of the leakage from blood vessels in response to micro-injection of VEGFA164 but not PBS. Glass capillary was filled with 1 μl VEGFA164 (1 μg/μl) and Alexa633 dye (cyan); about 0.1 μl was injected. Green, 2000 kDa FITC-Dextran. Red, 70 kDa TRITC-Ficoll. Images were acquired 5 min before and 30 min after injection. Bar, 50 μm

for atraumatic intradermal injection of small volumes (~0.1 μl) of recombinant proteins. Intradermal injection of VEGFA164, but not PBS, resulted in a rapid and transient induction of vascular leakage (Fig. 1b, see Supplementary Movie 1), in agreement with its established function[28].

**Blood vessel classification**. Different vessel types; arterioles, capillaries, venules, are known to respond differently to various stimuli[7,9,11]. Vessel types can be classified by their diameter: the diameter of arteries and veins would typically be >50 μm, arterioles and venules ≥10 μm, capillaries ≤10 μm[29]. The distribution of vessel types in the ear dermis microvasculature based on these parameters was 9% arterioles, 75% capillaries, and 16% venules (Supplementary Fig. 1), i.e., similar to previously published data[29]. The dermal vasculature has a largely stochastic arrangement making the distinction of different vessels of similar caliber, i.e., arterioles and venules, difficult. To initially distinguish between arterioles and venules, the progressive distribution of 2000 kDa FITC-Dextran through the vasculature was followed after bolus injection (Fig. 2a; heatmap classification in Fig. 2b, Supplementary Movie 2). Thus, the bolus first reached the arteries and arterioles (in 10 s) followed by capillaries and finally venules and veins (Fig. 2c). Another distinguishing hallmark of different vessel types is their different blood flow velocities. Flow estimations can be made from the velocity of RBC movement[25]. Vessel diameter measurement and dye bolus injection with subsequent imaging using XT (X dimension over time) line scanning acquisition identified distinct flow velocity values for arterioles (>100 μm/sec) and venules (<100 μm/s) (Fig. 2d). Similar to venules, capillaries exclusively displayed flow velocities of <100 μm/s. Thus, through vessel diameter and flow velocity, different vessel types could be identified (Fig. 2a–d).

It is well established that arteries and veins differ with regard to their capacity to respond to vessel agonists, e.g., with increased leakage[9]. Whether this difference in vascular barrier properties is related to the expression of TJ proteins has not been addressed. Immunostaining of the ear dermis for the TJ marker Claudin5 (Cldn5)[30] revealed Cldn5-positive regions of postarterial capillaries, in addition to arteries and arterioles (Fig. 2e). A typical postarterial capillary stretch would be positive for Cldn5 but gradually lose Cldn5 immunostaining in subsequent segments, while maintaining expression of the vascular marker Isolectin B4 (Fig. 2e; quantification in Fig. 2f). A transgenic mouse C57BL/6 strain expressing E (enhanced) GFP under the control of the Cldn5 promoter, denoted *Cldn5*(BAC)-GFP, allowed direct detection of *Cldn5* promoter activity and thereby, Cldn5 expression in time-lapse imaging. The *Cldn5*(BAC)-GFP mouse showed GFP expression in arteries, arterioles, but not in venules and veins in the adult ear dermis (Fig. 2g). Moreover, GFP expression was detected in postarterial capillaries (denoted positive capillaries in Fig. 2g), and in capillary stretches that showed a mixed pattern of GFP expression (mixed). There were also capillary stretches without apparent GFP expression (negative capillaries). Therefore, the pattern of GFP expression was in good agreement with the results of immunostaining for the endogenous Cldn5 (Fig. 2e). By following the distribution of a 2000 kDa TRITC-Dextran bolus in the *Cldn5*(BAC)-GFP ear dermis vasculature, the hierarchy of the different categories of capillaries could be mapped (Fig. 2h). GFP-positive arterioles were followed by GFP-positive and mixed capillaries, leading on to GFP-negative capillaries and venules.

Flow velocities in arterioles and venules in the ear dermis were found to be similar between wild-type and *Cldn5*(BAC)-GFP mice (Fig. 2d, i). Analysis of the different capillary categories in the *Cldn5*(BAC)-GFP mouse showed a tendency for faster flow

but still <100 μm/s for GFP-positive capillaries (i.e., those covered by Cldn5) whereas the mixed type and GFP-negative capillaries displayed even lower velocities (Fig. 2i and inset). Thus, in addition to vessel diameter and RBC velocity, vessel categories, as well as capillary subtypes, could be discerned based on the pattern of Cldn5 expression.

**RVDM analysis to measure dynamic changes in RBC velocity**. To follow rapid changes in permeability in different vessel types following intradermal stimulation, a large area of observation was necessary as we could not predict which vessels would respond to the stimulus. Moreover, to obtain reliable quantifications, we wished to measure changes in more than one vessel for each injection. Blood flow velocity is an important tool for distinguishing vessel types. However, techniques to measure flow velocity such as XT acquisition and fast acquisition in *X* and *Y* dimensions over time (fXYT) do not allow blood flow measurements when imaging large fields of view at slower frame rates (1–5 s/frame), i.e., sXYT acquisition. The RVFS method for blood flow analysis developed by Kamoun[25] exploits fluorescent tracing by following transplanted RBCs. RVFS thus allows analysis of flow velocity over a large area, but relies on scanning the field of view at different angles to allow analysis of vessels with different flow directions. This however interferes considerably with the required acquisition rate making it unsuitable for imaging of vascular dynamics following delivery of vessel agonists such as VEGFA.

We therefore developed an image analysis method, RVDM, that allows flow velocity measurements of essentially all vessels within a large field of view from a single frame of a time-lapse movie. When acquired using sXYT imaging, RBC images elongate or shrink depending on the relationship between the speed of laser scanning, blood flow velocity and flow angle, with firm reproducibility (Fig. 3a, Supplementary Fig. 2). The dimensions of the distorted RBC image ($X_s$; horizontal, $Y_s$; vertical), combined with the RBC residence time within the scan field ($T$), may then be used to determine the velocity of individual RBCs, and thus flow velocity (Fig. 3b, Supplementary Fig. 2 and Methods). As shown in Fig. 3c, comparisons of blood flow measurements from sXYT acquisition combined with RVDM analysis, with fXYT and XT imaging in various vessel types resulted in very similar estimations of RBC velocity at all speeds above which RBC distortion occurs using the set acquisition parameters (flow > ~20 μm/s). Slower flow velocities were calculated by tracking single RBCs in each frame. To demonstrate the ability of RVDM to easily calculate the flow velocities of vessels in a large vascular network it was applied to the image shown in Supplementary Fig. 1. The individual images that make up this broad 3D network were subject to RVDM analysis and the calculated velocities translated into a velocity map (Supplementary Fig. 3). This clearly shows the usefulness of RVDM in calculation of the velocities of all vessel types within a network using simple imaging parameters. Similar to previous reports, the map shows that vessel velocities may change markedly as vessels intersect and bifurcate[25]. Thus, RVDM provides an accurate quantification of blood flow velocity, and due to its compatibility with simple sXYT imaging, allows blood flow measurement alongside other dynamic vascular events.

**Dynamics of vascular leakage**. While vascular leakage is recognized as an essential feature of vessel physiology and pathology[31], the precise location and dynamics of leakage in response to different stimuli remain to be discerned. We applied MPLSM with subsequent RVDM analysis on the ear skin. Intradermal injection of VEGFA164 in the wild-type C57BL/6 ear resulted in leakage of tail vein administered FITC-conjugated 2000 kDa Dextran and

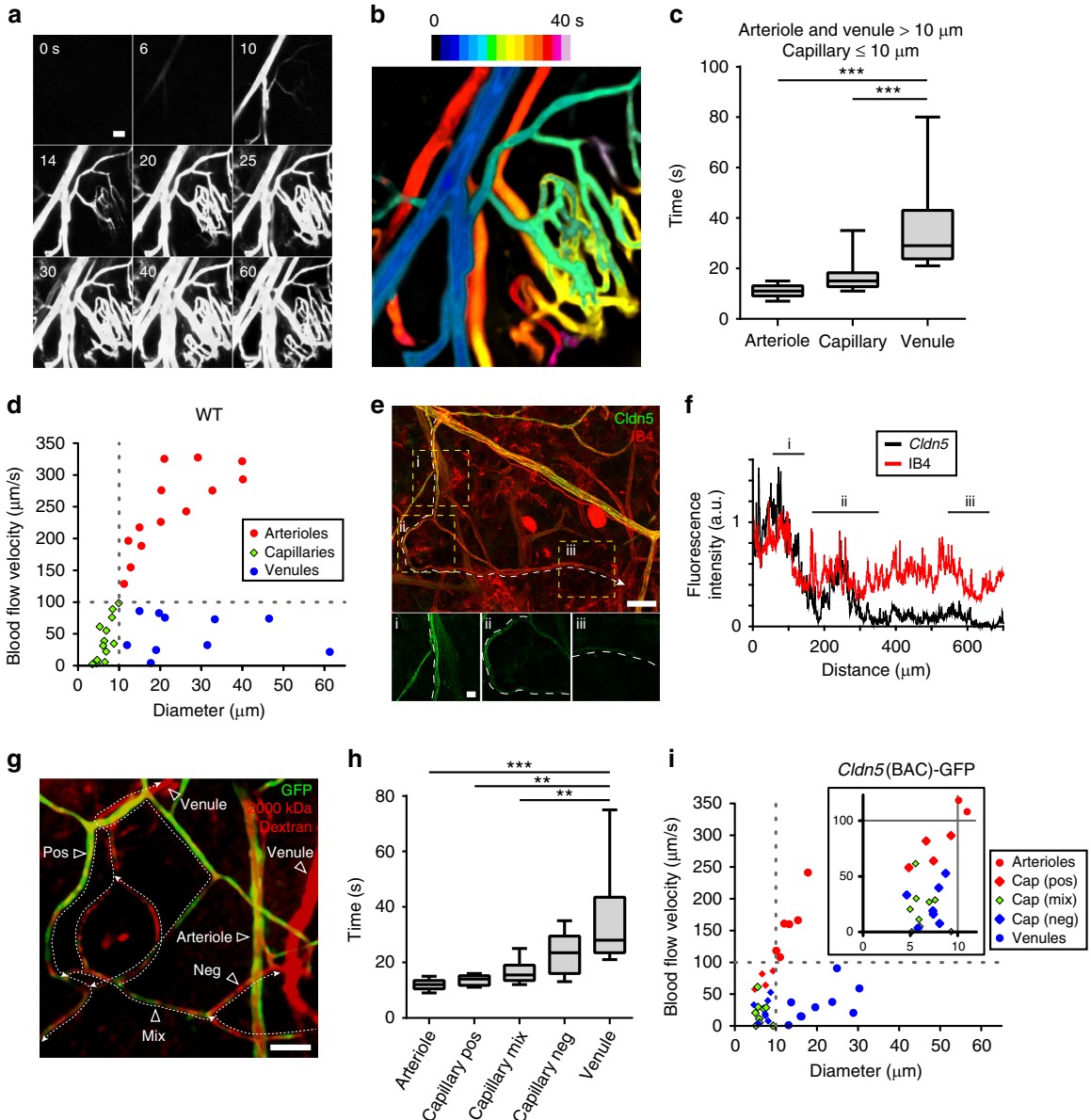

**Fig. 2** Identification of vessel types in the ear dermis. **a** Distribution kinetics (frames shown at 0 up to 60 s after injection) of 2000 kDa FITC-Dextran injected as a bolus in the tail vein. Single-photon time-lapse imaging (0.5–1 s/frame) was performed without pinhole to capture a wide focal depth. Bar, 50 μm. **b** Heatmap of the distribution kinetics of 2000 kDa FITC-Dextran, visualizing arterioles (dark blue), capillaries (green to yellow) and venules (yellow to dark red). **c** Box plot showing distribution kinetics of 2000 kDa FITC-Dextran for different vessel types, defined as arterioles and venules (>10 μm) and capillaries (≤10 μm), with median (center line), 25th and 75th percentiles (box bounds), and whiskers (maximum and minimum data point) indicated. $n = 7$ mice. Tukey–Kramer test; ***$p < 0.001$. **d** RBC velocity in arterioles, capillaries, and venules in wild-type C57BL/6 ear dermis. $n = 7$ mice with >20 RBCs measured/vessel-type (10–14 vessels of each type). **e** Ear dermis vasculature, merged immunostaining for Cldn5 (green), and Isolectin B4 (IB4; red). Boxes labeled i–iii with white dashed vessel outlines are shown as enlarged images below, illustrating consecutive capillary segments that gradually lose Cldn5 expression. Bar, 50 μm and 10 μm in lower panels. **f** Line plot of Cldn5 and IB4 fluorescence intensities along capillary segments in **e**. **g** Cldn5(BAC)-GFP mouse ear dermis after injection of 2000 kDa TRITC-Dextran, with different vessel types indicated. Note that venules and certain capillary segments show circulating TRITC-Dextran (red) but do not express GFP (green). Capillaries with GFP expression (positive; pos), with mixed expression (mix) and no GFP expression (negative; neg) are indicated. Dashed lines with arrowheads show the direction of blood flow. Bar, 50 μm. **h** Box plot showing distribution kinetics of 2000 kDa TRITC-Dextran to different vessel types with median (center line), 25th and 75th percentiles (box bounds) and whiskers (maximum and minimum data points) indicated. Note that dextran reached the GFP-expressing capillaries before non-expressing capillaries. $n = 3$ mice (independent biological repeats). Tukey–Kramer test; **$p < 0.01$, ***$p < 0.001$. **i** RBC velocity in arterioles, venules, and different capillaries, positive (pos), mixed (mix), and negative (neg), for GFP expression in Cldn5(BAC)-GFP ear dermis. $n = 3$ mice with >20 measurements/vessel-type

TRITC-conjugated 400 kDa Ficoll (Fig. 4a). Leakage occurred at distinct points (denoted leakage points, see Fig. 4a) that were established at intervals along the vessel length. Leakage points were detected on capillaries and venules, as distinguished by flow velocity, quantified by RVDM, and vessel diameter (Fig. 4a, high

magnification to the right). In the Cldn5(BAC)-GFP mouse the extravascular spread of 400 kDa TRITC-Ficoll also occurred from capillaries and veins (Fig. 4b, c). Quantification of leakage points in the various vessel types showed no induction of leakage from GFP-positive, Cldn5-expressing arteries and capillary segments,

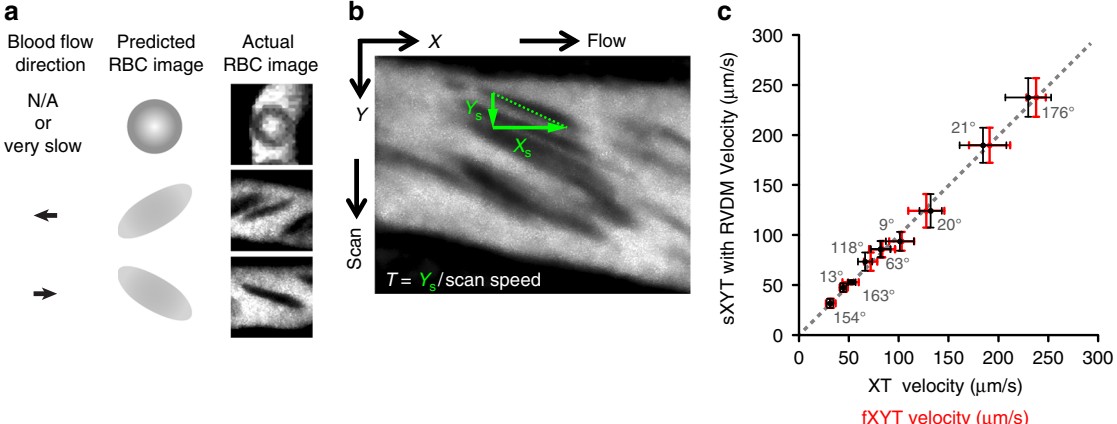

**Fig. 3** RVDM principles and verification. **a** RBC image is influenced by the speed of laser scanning relative to blood flow velocity and direction resulting in distorted RBC dimensions. **b** sXYT imaging produces distorted RBC images from which RBC dimensions $X_s$ and $Y_s$ and RBC residence time are measured within the scan field ($T$), thus allowing RBC velocity quantification (see Supplementary Fig. 2 and Methods). **c** Comparison of RBC velocities estimated from sXYT combined with RVDM, XT, and fXYT image acquisition in the same vessels. Flow angles are indicated for each measurement. $n = 3$ mice with more than 20 measurements/vessel

while capillary segments with mixed or no GFP-positivity, as well as venules, leaked in response to VEGFA164 to a similar extent (Fig. 4d).

sXYT image acquisition was next used to determine leakage dynamics over time in mice with circulating 2000 kDa FITC-Dextran and 70 kDa TRITC-Ficoll (Fig. 4e–h). Kymograph images (Fig. 4e lower panel) were generated of vessel cross-sections at points of fluorescent dextran leakage after VEGFA164 injection (Fig. 4e, upper panels). Analysis of fluorescence changes just outside the vessel wall in the kymographs was used to determine the leakage time course (Fig. 4f–h).

Within a 100 μm-stretch from the point of injection, the lag period was reproducible (Supplementary Fig. 4). Only leakage sites between 50 and 100 μm from the injection site were subsequently recorded to control for diffusion of the injected substances, and to avoid variability due to high ligand concentrations and changes in interstitial pressure close to the injection site. As shown in Fig. 4f, g, leakage of 70 kDa TRITC-Ficoll was initiated in <2 min (i.e., the lag period) after intradermal injection of VEGFA164. The lag period was slightly longer, although not significantly, for the 2000 kDa FITC-Dextran. Moreover, venules and capillaries showed similar responses to VEGFA164.

After leakage initiation, fluorescence intensity outside of the vessel increased to its maximum, signifying closure of the leakage points. Leakage continued for 6–12 min (leakage duration) dependent on the vessel-type and the size of the dextran (Fig. 4f, h). The duration of leakage was the longest (~12 min) for 70 kDa TRITC-Ficoll leaking from capillaries whilst the shortest leakage duration occurred in venules, where leakage of the larger dextran stopped after 6 min.

**Dynamics of vessel dilation and blood flow**. VEGFA is known to induce vasodilation[3], however the kinetics and engagement of different vessel types have remained unclear. Injection of VEGFA164 caused marked changes of vessel diameter in the ear dermis (Fig. 5a and Supplementary Movie 3). To gain insights into the dynamics of vasodilation, we used sXYT imaging with subsequent RVDM and morphology analysis to separate vessel types and determine vessel dynamics. Venules and capillaries meanwhile responded with vessel dilation, with similar kinetics as for vascular leakage. While venules changed their diameter less than 1.7-fold over the time period of observation, capillary

dilation was marked, with a threefold increase over the first 5 min after VEGFA164 injection (Fig. 5b). Thirty minutes after stimulation both venules and capillaries remained dilated.

Blood flow dynamics before and after injection of VEGFA164 was followed using sXYT imaging and RVDM analysis. In response to VEGFA164, the flow velocity decreased significantly in venules (Fig. 5c). The reduction in RBC velocity was even more marked in capillaries. Still, after 30 min, although slowly returning back to resting conditions, the capillary flow velocity was lower than before injection of VEGFA164.

Comparing the lag periods of the different vascular responses to VEGFA showed that leakage and dilation could not be kinetically separated (Fig. 5d). There was a trend that changes in RBC velocity in response to VEGFA164 occurred subsequent to the concurrent induction of leakage and vasodilation.

## Discussion

Here, we present a MPLSM-based non-invasive intradermal stimulation technique and image analysis tool, RVDM (relative velocity, direction, and morphology) for the quantitative assessment of simultaneously occurring, highly dynamic events such as vascular permeability, and changes in vasotone and blood flow. Non-invasive intradermal stimulation may also be of use for the study of, for example, inflammatory settings and leukocyte dynamics. Meanwhile, the procedure for estimation of velocity dynamics could be adopted to recordings, e.g., of the movement of different blood cells or the response of abnormal tumor vessels to various drugs that inhibit signaling in specific pathways or to test effects of clinically applied therapies.

We moreover provide information on the identity of vessels that respond to VEGFA. Vessels in the dermis were defined as capillaries and postcapillary venules, respectively, based on their anatomical location in the vasculature, width (≤10 μm for capillaries) and RBC velocity parameters. Postarterial capillaries were covered by Cldn5, followed by a segment with a salt-and-pepper pattern with variable but diminishing expression levels. The subsequent vessel segments (prevenular and postcapillary) showed very low if any Cldn5 expression. Cldn5-deficient capillaries as well as postcapillary venules leaked in response to VEGFA164. At present, there is no strict molecular or morphological criteria to distinguish prevenular capillaries from postcapillary venules in the skin. However, the designation of the leakage-permissive vessel segment (defined as ≤10 μm, low-flow and Cldn5-deficient) as

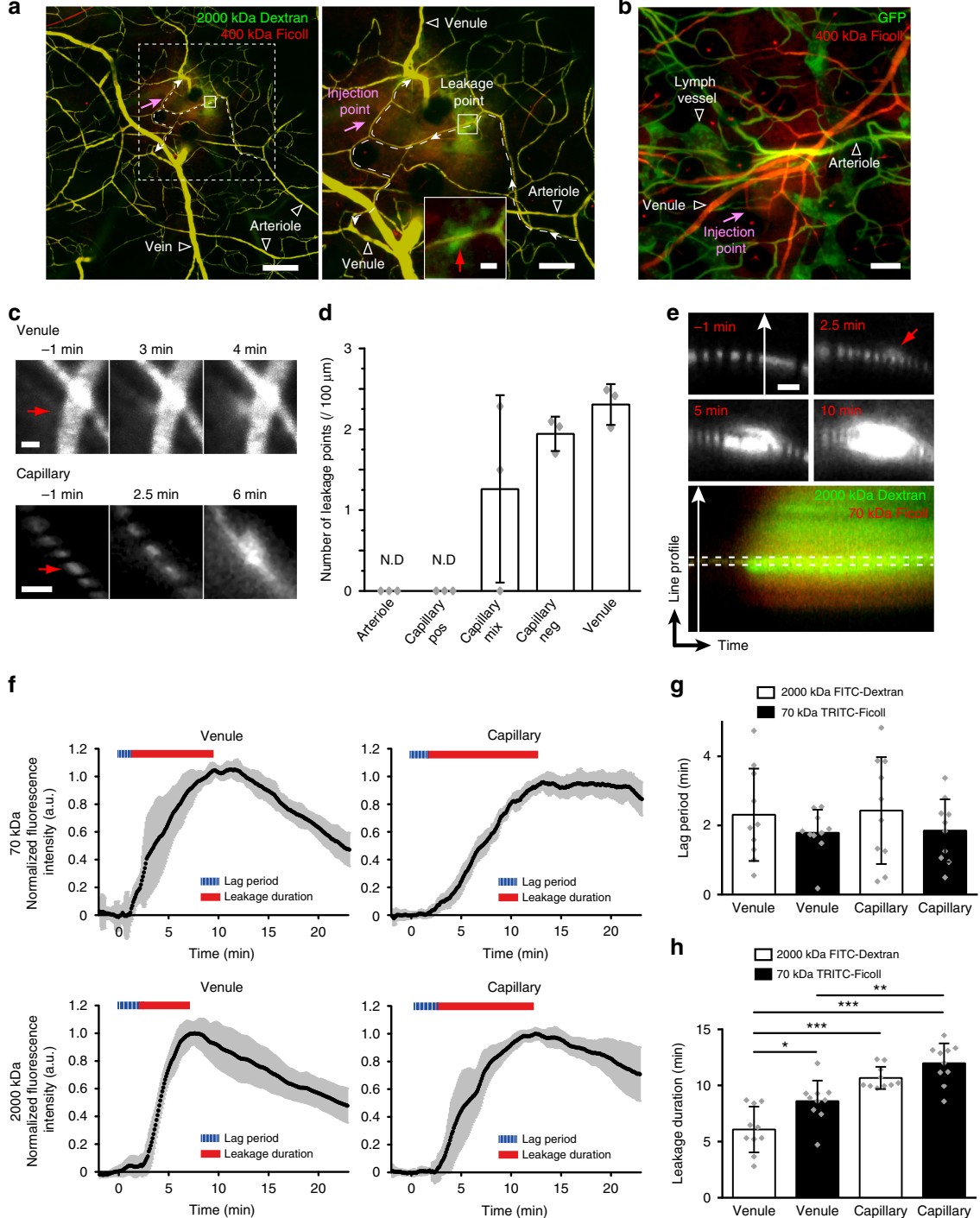

**Fig. 4** Dynamics of VEGFA-induced vascular leakage. **a** Point of leakage of circulating 2000 kDa FITC-Dextran and 400 kDa TRITC-Ficoll in a large observation area (left) and zoom-in (right). Flow direction (arrows) and leakage point in a capillary (dashed line), with leakage point boxed (solid line). Right panel, magnification of dashed box area. Capillary leakage point indicated by red arrow. Bar, 200 (left), 100 (right) and 10 μm (inset). **b** Leakage of 400 kDa Ficoll in response to VEGFA164 in the ear dermis of *Cldn5*(BAC)-GFP mice. Bar, 100 μm. **c** Leakage points (red arrow) in *Cldn5*(BAC)-GFP mouse dermis, in venule (upper) and capillary (lower), showing progressive leakage over time of 400 kDa TRITC-Ficoll in response to VEGFA164. Bar, 10 μm. **d** Quantification of leakage points/vessel length in the *Cldn5*(BAC)-GFP ear dermis. Capillaries with GFP expression (positive; pos), mixed (mix) or no GFP (negative; neg) expression were analyzed. Note that Cldn5-expressing capillaries do not leak. N.D.; not detected. $n = 3$ mice with 6 vessels of each type analyzed/mouse. **e** Representative leakage point captured by sXYT imaging at −1, 2.5, and 10 min after injection of VEGFA164 (upper and middle panels). Kymograph image (lower panel) for analysis of leakage dynamics of 2000 kDa FITC-Dextran and 70 kDa TRITC-Ficoll was made from the point indicated by the white arrow line in upper panels. Red arrow indicates the leakage point. Bar, 10 μm. **f** Leakage kinetics from venules (left) and capillaries (right) of 70 kDa TRITC-Ficoll (upper) and 2000 kDa FITC-Dextran (lower) with lag periods and leakage durations indicated. Gray area around solid black line indicates variability (S.D.). $n = 7$ mice with 10 capillaries and 10 venules analyzed. **g** Quantification of lag periods (min) for the different conditions. $n = 7$ mice and 10 vessels/condition as in **f**. No significant difference by Tukey–Kramer test. **h** Quantification of leakage duration (min) for the different conditions. $n = 7$ mice and 10 vessels/condition as in **f**. Tukey–Kramer test; *$p < 0.05$, **$p < 0.01$, ***$p < 0.001$

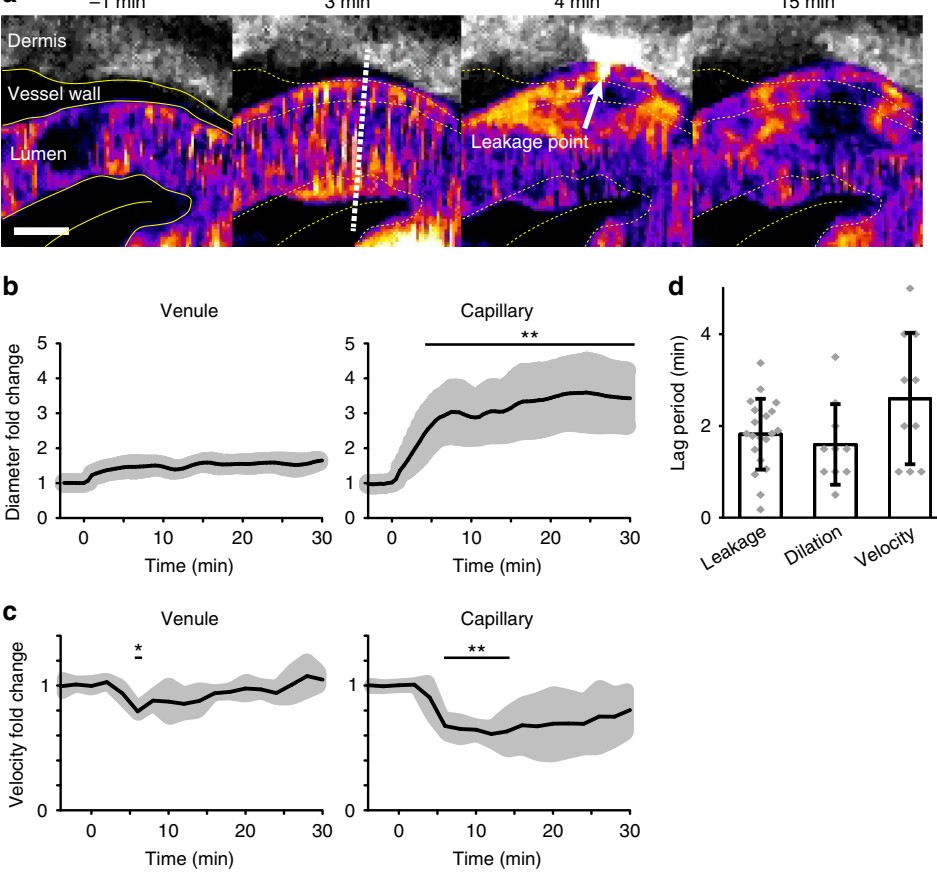

**Fig. 5** Vasodilation and RBC velocity dynamics in response to VEGFA. **a** Illustration of vasodilation at a leakage point induced in response to VEGFA164 administration. Colors: 2000 kDa FITC-Dextran in lumen; pseudocolor, 2000 kDa FITC-Dextran in dermis; white. White dotted line marks the position used for measurement of vessel diameter dynamics by FWHM at a leakage point (arrow). Bar, 5 μm. **b** Kinetics of vasodilation at different time points after administration of VEGFA164, in venules (left) and capillaries (right). Gray area around solid black line indicates variability (S.D.) $n = 3$ mice (individual biological repeats) with 5 vessels of each kind analyzed. Student's $t$-test; $^{**}p < 0.01$. **c** RBC velocity dynamics in response to VEGFA164 in venules (left) and capillaries (right). Gray area around solid black line indicates variability (S.D.) $n = 3$ mice and 5 vessels of each kind analyzed with more than 20 RBCs/time point. Student's $t$-test; $^{*}p < 0.05$, $^{**}p < 0.01$. **d** Comparison of lag periods for the different responses to VEGFA164, leakage, vasodilation, and velocity changes. $n = 20$ vessels for leakage analysis in 7 mice and $n = 10$ vessels in 3 mice (combined capillaries and venules) for dilation and velocity analyses as in **b** and **c** panels, respectively. No significant difference by the Tukey–Kramer test

"capillary" rests on its distinction from the following venular segment with regard to VEGFA-induced leakage duration, and changes in blood flow and vasotone (Figs. 4 and 5).

The leakage from Cldn5-deficient capillaries and venules may not be explained solely on the absence of Cldn5. Constitutive deficiency in Cldn5 expression results in size-selective loosening of the blood–brain barrier and death immediately following birth[32]. The cause of the post-partum death of Cldn5-deficient pups has not been clarified and may possibly include disturbances also in the peripheral vasculature. Whether there are organ-specific differences in the distribution of Cldn5 remains to be studied. We also cannot exclude that there are low levels of Cldn5 expression in vessels we denoted as deficient, out of the range of detection using MPLSM and the *Cldn5* promoter-driven GFP model applied here.

RVDM analysis provides the ability to utilize native, unlabeled RBCs to calculate the flow velocity of essentially all vessels within a large field of view using simple imaging parameters. Detailed instructions describing how to utilize the RVDM analysis are given in the walk-through and trouble-shooting guide (Supplementary Note 1), however, important aspects should be taken into consideration. Firstly, it should be noted that we cannot rule out that variation in RBC dimensions may exist between species,

strains or in certain pathological situations. The walk-through and trouble-shooting guide (Supplementary Note 1) details how to calculate these dimensions. Furthermore, caution should be taken when studying disease situations where RBC dynamics might differ from the normal situation. RBCs are known to form an ellipsoidal shape when under shear stress, which can undergo a 'tank-treading' motion[33]. However, diseases such as various anemias may alter normal RBC behavior, resulting in a more chaotic motion and higher baseline error.

RVDM velocity calculations rely on the relationship between the relative speed and angle of blood flow and laser scanning. Because of this, RVDM can be used to calculate blood velocities in vessels flowing in any direction with a single image. It should be noted that certain angle relationships, for example those flowing against the direction of image capture, may hinder calculation of accurate velocities due to the shrinkage, rather than elongation, of RBC images. To minimize the effects of such scenarios, however, RVDM analysis rejects unreliable results based on strict parameters to ensure accurate velocity values.

The imaging parameters used in this study allow the calculation of blood flow velocity in vessels of any orientation in the ear dermis, where velocities reach 250 μm/s (Fig. 3c). Other vascular beds may present speed relationships that for example result in

endless RBC images when the scanning speed matches RBC velocity, or no or very small RBC images due to very fast flow velocity. Nevertheless, such situations can be overcome by altering the imaging parameters to give faster or slower scan speeds. Furthermore, some vessel types and their position within the vascular network may have higher hematocrits. While the close packing of RBCs can impede analysis, the automatic rejection of improbable velocity values within the RVDM calculations ensures that no such scenarios will be included in final results. The deviation of the estimated flow velocity for individual vessels using RVDM was within the range of 5–15%, similar to the deviation seen with the established techniques XT and fXYT (Fig. 3c). These deviations, which were estimated based on repeated measurements at a single point, may be technical or biological, in the latter case due to rapid changes in blood flow. Thus, vessel velocity can be measured reliably using RVDM as long as the flow velocity is compatible with the scan speed and provided that RBCs indeed are flowing through the vessel. If there is no flow, time-lapse acquisition allows temporarily empty vessels to be analyzed. Moreover, the error in blood flow measurements using RVDM increases when flow angles are near $\theta = 0$, 90, 180, and 270°. On the other hand, the error for values of $\theta = 0$–15°, 75–105°, 165–195°, 255–285°, and 345–360° is minimized as these flow angles are subject to a more stringent selection (see Methods section Relative velocity, diameter, and morphology (RVDM)). In such situations, however, other sections of a vessel segment may be chosen as single vessels within a given field are commonly not straight and segments can readily be found that possess a more favorable flow angle. Therefore, while RVDM relies on user input, it does not rely on user accuracy and provides unbiased quantification of blood flow velocity similar to fully automated methods, the implementation of which are currently unsuitable for complex biological processes where conditions continuously change. Given careful imaging and selection of appropriate imaging parameters, RVDM provides a powerful means of calculating blood flow velocity simultaneously with other dynamic events.

Vascular permeability to small molecules (<70 kDa[4]) occurs in the static condition by sieving through established AJs[5]. Passage of larger plasma proteins (fibrinogen, immunoglobulins) requires that AJs dismantle in response to agents such as VEGFA and inflammatory cytokines[11,34]. The dismantling involves internalization of VE-cadherin, creating a dynamic passage for larger molecules between endothelial cells. Passage of leukocytes and immune cells may require additional parameters yet to be identified[35]. Vascular leakage has been studied in different pathology settings, mainly in cancer. A hallmark of cancer vessels is their chronic leakiness, contributing to a chaotic tumor microenvironment, tumor edema, and disease progression[1]. As such, a more in depth understanding of the mechanisms contributing to such leakiness may enhance therapeutic strategies.

Using non-invasive intradermal stimulation, we show that VEGFA164-induced leakage in the healthy skin occurs with very strict dynamics. About 2 min after the intradermal injection of VEGFA, leakage was initiated. During this time span, junctions dismantled allowing leakage of molecules as large as 2000 kDa dextran. The highly reproducible lag period also involved diffusion of VEGFA to reach the endothelial cell and initiate VEGF receptor signaling. It is conceivable that an endogenous VEGFA source, such as an inflammatory cell, would be located very close or on the vessel, shortening the lag phase considerably. After about 6–12 min dependent on the vessel-type and the dimensions of the leakage tracer, the leakage gradually dropped and vessel integrity became reestablished. While opening of AJs involves several different signal transduction cascades such as VEGF/VEGFR2-induced Src family kinase signaling, and endothelial nitric oxide synthase (eNOS)-signaling[36,37], it is unclear what

mechanisms operate to reestablish AJs. Of note, trans-vessel passage may involve additional mechanisms such as VEGFA-regulated vesiculo/vacuolar trafficking, potentially occurring at endothelial junctions[21].

Vasodilation was recorded as a marked response to VEGFA164, in particular in capillaries, which increased their diameter three-fold (Fig. 5a, b). This effect of the locally active paracrine factor VEGFA is in contrast to systemic neurogenic stimulation (whisker stimulation in rodents), which results in arterial dilation accompanied by increased RBC velocity[27], illustrating that distinct mechanisms of vasodilation may exist. During the observation period, we recorded only partial reversion of vessel diameter to pre-stimulation dimensions. Therefore, vasoconstriction does not appear to be essential in reestablishment of vessel integrity and closure of AJs, at least not in capillaries. Dilation of capillaries in response to VEGFA was accompanied by a marked decrease in RBC velocity, identified by application of RVDM. In comparison, RBC velocity and vasotone in venules was only moderately affected (Fig. 5a). We do not exclude that the change in RBC velocity was related to the marked vasodilation, or that the estimation of the velocity change was influenced by the different properties of the dilated vessels. However, the movies captured of dilating, leaky vessels clearly demonstrate a marked decrease in RBC velocities. We moreover agree with the notion that the changes in RBC velocities may not directly translate to changes in other blood constituents. In summary, the hitherto unknown molecular mechanisms underlying the very strong effect of VEGFA preferentially on capillary vasotone and RBC velocity, and the consequence for the microenvironment, will be addressed in future studies using the methodologies presented here.

## Methods

**Animals**. Wild-type female C57BL/6J mice age 10–18 weeks were used, unless otherwise indicated, for live imaging of the ear vasculature. In certain cases, C57BL/6J mice (also females and 10–18 weeks), expressing enhanced green fluorescent protein (EGFP) under the control of the mouse Claudin5 promoter, denoted *Cldn5* (BAC)-GFP, were used. In vivo animal experiments were carried out in strict accordance with the ethical permit provided by the Committee on the Ethics of Animal Experiments of the University of Uppsala (permit no C119/13). After injection in the ear dermis and in vivo imaging, mice were killed. Care was taken to avoid unnecessary suffering for the animal. Each experiment was conducted on tissue from at least three animals on at least three different experimental days. Mice were not randomized for the live imaging. Tissue samples were not blinded. Sample size (number of vessels observed in number of mice) were chosen to ensure reproducibility and allow stringent statistical analysis. See figure legends for information on sample sizes. All experimental conditions were reproduced with at least 3, most often up to 7 mice at different time points, representing individual biological repeats.

**Growth factors and antibodies**. Recombinant mouse VEGFA164 (493-MV/CF; R&D Systems) and canine VEGFA164 (a kind gift of Dr. Kurt Ballmer-Hofer, Paul Scherrer Institut, Villigen, Switzerland), which share 99% amino acid sequence identity, were used at a concentration of 1 μg/μl for intradermal injection. Repeated tests in this and other studies[37] showed no difference in efficiency to induce VEGFR2 activation and vascular leakage[37]. Approximately 0.1 μl (i.e., 100 ng VEGFA) was injected in each experiment.

Anti-Claudin5 antibody (1:100 dilution; cat no 341600, Thermo-Fisher; validated by Nitta et al.[32]) and Isolectin GS-IB$_4$ From *Griffonia simplicifolia*, Alexa Fluor™ 647 Conjugate (1:400 dilution; cat no I32450, Thermo-Fisher) were used where stated.

**Tail vein cannulation and glass capillary**. A cannula made from a 30-gauge syringe needle tip and thin silicone tubing was inserted into the tail vein to administer fluorescent conjugates. Sub-micron glass capillaries with a tip diameter below 1 μm were used for intradermal injections. Capillaries were manufactured in-house using a handmade glass puller or micropipette puller (P-30, Sutter instrument) from borosilicate glass microtubule (BF150-120-10, Sutter instrument).

**Immunofluorescent staining**. Ears were removed and fixed in 4% paraformaldehyde (PFA) for 2 h at room temperature. After dissection, the tissue was post-fixed in 100% methanol at −20 °C for 10 min and blocked overnight at 4 °C in Tris-buffered saline (TBS) with 5% (w/v) nonfat dry milk and 0.2% Triton X-100. Samples were incubated overnight with primary antibody in blocking reagent,

followed by washing several times in TBS with 0.2% Triton X-100 and incubation with appropriate secondary antibody for 2 h at room temperature in blocking buffer. For isolectin B4 staining, samples were subsequently fixed for 10 min with 4% PFA at room temperature before washing several times in PBlec (1 mM MgCl$_2$, 1 mM CaCl$_2$, 0.1 mM MnCl$_2$ and 1% Triton X-100 in PBS). Alexa647 conjugated isolectin B4 (132450, Thermo-Fisher) was then incubated with samples in PBlec for 2 h before washing several times and mounting in fluorescent mounting medium (DAKO). Images were acquired using a Leica SP8 confocal microscope.

**Single or multi-photon imaging**. Mice were anesthetized by intraperitoneal injection of Ketamine-Xylazine (62.5 mg/kg Ketamine and 8.3 mg/kg Xylazine). After a surgical level of anesthesia had been reached, the mouse was placed on a plastic plate with a heating pad covered with a cotton pad to maintain a body temperature of ~37.5 °C, recorded using a rectal probe. The depth of anesthesia was continuously monitored by testing the animal's reflexes (e.g., pedal or eye blink reflex) during the experiment.

In vivo imaging of blood vessels was performed using a single or multi-photon scanning microscope (Zeiss LSM710), equipped with a MaiTai HP Ti:Sapphire laser (SpectraPhysics) and DPSS Lasers, and a high N.A water-immersion objective lens (CFI75 Apochromat 25xW MP N.A.1.1, Nikon or W Plan-Apochromat ×20 N.A.1.0; Zeiss). Tissues were excited using a 960–1040 nm laser with a power of 0.5–20 mW under the objective lens, to avoid photo-damage. The emitted light was filtered to collect green and red or far-red fluorescent signals and second harmonic generation signals (BP475/42, BP525/50, BP641/75, FF495-Di03, Di02-R594, Semrock). Single-photon excitation was used for time-lapse imaging of dye bolus injections and three-color acquisition. The fluorescent signals were separated into green (490–550 nm), red (580–620 nm) and far-red fluorescence (640–680 nm) signals by grating methods.

To visualize the vasculature and follow dynamic changes, 100 µl solutions containing 50 mg/ml of 2000 kDa TRITC-Dextran (Cldn5(BAC)-GFP) or a mixture of 2000 kDa FITC-Dextran and 70 kDa TRITC-Ficoll or 400 kDa TRITC-Ficoll in PBS (Sigma or Thermo-Fisher Scientific), were injected into the circulation via the tail vein cannula. Z-stack images were taken of the same visual field as observed by time-lapse imaging, before and after injections, to determine distribution of leakage points and vessel diameter. Slow time-lapse imaging (sXYT) was used to image blood vessel responses to vascular ligands with or without Alexa633 to trace injections using the sub-micron glass capillary.

**Scanning speed and image size**. Images were taken by one or two-photon microscopy with the following scan speeds and image size; ~1 ms/line, 1024 pixel/ line and under 0.1 µm/pixel for line scanning (XT), 20–100 ms/frame and under 0.2 µm/pixel for fast frame scan (fXYT), 0.5–1 s/frame for dye bolus analyses, 1–5 s/frame and over 1024 × 1024 pixel/frame and 0.2 µm/pixel for slow time-lapse (sXYT) and Z-stack images, and 1–10 s/frame and under 0.5 µm/pixel for tile scan Z-stack images.

**Vessel diameter**. The diameter of arterials, capillaries and venules were calculated using the full-width half-maximum (FWHM) method[38], to provide accurate measurements regardless of fluorescent signal intensity and background noise levels. Briefly, the FWHM value was found following the fitting of Gaussian distribution to the intensity values of a line drawn perpendicular to the direction of blood flow and at least twice the vessel width. Measurement accuracy was ensured following calibration with 3 µm fluorescent beads.

**Relative velocity, diameter and morphology (RVDM)**. RBCs under very slow to no flow were measured to have a diameter of 5.3 µm by FWHM (Supplementary Figs. 2a, c). However, RBCs subject to shear stress change their shape, with those under flow becoming more ellipsoidal[33]. In addition, RBCs under flow can oscillate, due to the turbulent nature of blood flow, and also rotate around their axis parallel to flow, resulting in variations in the observed orthogonal axis[33]. To account for this variation and obtain a baseline image to which distorted RBC images could be compared, RBCs were imaged using fXYT (~20 ms/frame) to avoid the distortion caused by slow scan speeds. Subsequently their parallel and orthogonal axes in relation to the flow direction were measured (length 2a and 2b, respectively; Supplementary Fig. 2b, c). From this, we could calculate the mean RBC dimensions under flow (2a = 4.8 µm, 2b = 3.6 µm; n > 100) and thus define the co-ordinates a and b (Supplementary Fig. 2c, d); note the greater variation in 2b due to RBC rotation around the axis parallel to flow. Using elliptical tangent formulae, flow angle compared to laser scan line (θ) and co-ordinates a and b from the parallel and orthogonal axes, we could then calculate co-ordinates c and d using Eqs. (1) and (2) (Supplementary Fig. 2e). The points c and c′ are defined as the co-ordinates of the tangent points formed with the laser scan line and d is the point horizontal to c and vertical to the RBC center (0), defined as the point where the axis between c and c′ meets the long axes parallel and orthogonal to flow. Co-ordinates c and d, and Eqs. (3) and (4) were then used to calculate mean RBC

lengths X ($X_m$) and Y ($Y_m$) between points c and c′ (Supplementary Fig. 2e)

$$\begin{pmatrix} c_x \\ c_y \end{pmatrix} = \begin{pmatrix} \frac{a^2 \tan\theta}{\sqrt{a^2 \tan^2\theta + b^2}} \\ \frac{b^2}{\sqrt{a^2 \tan^2\theta + b^2}} \end{pmatrix} \text{ (Supplementary Fig. 2d, e, point } c) \quad (1)$$

$$\begin{pmatrix} d_x \\ d_y \end{pmatrix} = \begin{pmatrix} \frac{\tan\theta\sqrt{a^2 \tan^2\theta + b^2}}{\tan^2\theta + 1} \\ \frac{\sqrt{a^2 \tan^2\theta + b^2}}{\tan^2\theta + 1} \end{pmatrix} \text{ (Supplementary Fig. 2e, point } d) \quad (2)$$

$$X_m = 2\sqrt{(d_x - c_x)^2 + (d_y - c_y)^2} \quad (3)$$

$$Y_m = 2\sqrt{(d_x)^2 + (d_y)^2} \quad (4)$$

Capturing images of moving RBCs will require fewer or more scan lines depending on the relative velocity and direction of blood flow and laser scan. Therefore, sXYT image acquisition results in distortion of RBC image proportional to blood flow velocity and direction and laser scan parameters (Supplementary Fig. 2f, g). Thus, by comparing the morphology of distorted RBC images, captured by sXYT, with the average baseline RBC shape, captured and calculated from fXYT imaging and Eqs. 1–4, we can calculate the distance travelled (red stippled line in Supplementary Fig. 2f) during RBC image capture and thus their velocity. Measurement of RBC dimensions X and Y during sXYT acquisition ($X_s$ and $Y_s$), and subtraction of $X_m$ and $Y_m$, provides the distance travelled along the x ($\Delta x$) and y ($\Delta y$) axes by RBCs under flow during their residence within the laser scan field (Supplementary Fig. 2h). RBC residence time (T) during image capture could be calculated by Eq. (5). Using Eqs. (6) and (7) flow velocities along the x- and y-axes ($V_x$ and $V_y$) could be calculated from $\Delta x$ and $\Delta y$ respectively. RBC velocity along the direction of flow could then be calculated independently from $V_x$ and $V_y$ using θ and cosine or sine transformation.

$$T = \frac{|Y_s|}{\text{Scan speed}} \quad (5)$$

$$V_x = \frac{|X_s - X_m|}{T} \quad (6)$$

$$V_y = \frac{|Y_s \mp Y_m|}{T} \quad (7)$$

To avoid error caused by abnormal RBC rotation and adjacent RBCs, RBC velocity values obtained from $V_x$ and $V_y$ were compared after their conversion to each other using tangent transformation and θ. Values between flow angles (θ) of 15–75° or 105–165° were rejected when differing by more than ±34.1%. Values of θ between 0–15°, 75–105°, and 165–180°, $V_y$ and $V_x$, respectively, offer inaccurate values and thus these angles were not subjected to this error check. To further decrease error, values were rejected that fell outside $2\sigma$ (θ = 15–75° and 105–165°) or a more stringent $1\sigma$ (θ = 0–15°, 75–105°, and 165–180°).

The software for performing RVDM analysis is provided in excel format as Supplementary Software 1. A "Walk-through and trouble-shooting guide" is provided as Supplementary Note 1.

**Statistical analyses**. Data are expressed as mean ± S.D. The principal statistical test was Students' t-tests, Mann–Whitney U tests (for non-Gaussian distributed data) or the Tukey–Kramer tests (multiple comparisons) as appropriate. p-values given in the text are from independent samples analyzed by two-tailed t-tests. For multiple comparisons, p-values were corrected using a procedure based on the Tukey–Kramer method. Normality of data was assessed using the Kolmogorov–Smirnov tests. All tests run were two-tailed. All statistical analyses were conducted using KyPlot or Excel software. A p-value <0.05 was considered statistically significant and significances were indicated as p < 0.05 (*), p < 0.01 (**), and p < 0.001 (***).

**Data availability**. The data that support the findings reported in this study are available from the corresponding authors upon reasonable request.

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

## Acknowledgements

We are grateful to Prof. Donald McDonald, UCSF, for valuable advice throughout the study and critical reading of the text. We also thank Dr. Lakshmi Venkatraman for critical reading and Ms Marie Hedlund for expert assistance with regard to animal experimentation. The Uppsala University BioVis imaging facility with Dr. Jeremy Adler is acknowledged for assistance during the initial stages of the study. This study was made possible through grants to L.C.-W. from the Swedish Research Council, the Swedish Cancer foundation, the Knut and Alice Wallenberg foundation (KAW) to L.C.-W. and C.B. (KAW 20150030). KAW also supported L.C.-W. with a Wallenberg Scholar grant. M.R. was supported by EMBO fellowship number ALTF 923–2016. N.H. was supported by Astellas Foundation for Research on Metabolic Disorders and The Uehara Memorial Foundation as a research fellowship.

## Author contributions

N.H. and M.R. performed all experiments and developed RVDM image analysis tool; M. S.-J. performed certain experiments; B.L. and C.B. provided unique reagents. Experiments were planned and interpreted and the manuscript assembled by N.H., M.R. and L. C.-W. All authors read and approved the manuscript.

## Additional information

**Competing interests:** The authors declare no competing interests.

