## [Peer Review File · Nature Communications]

Reviewer #1 (Remarks to the Author):

The authors present novel methodology for studying blood flow and vessel physiology intravitaly, over large regions of tissue. They study how various stimuli affect vessel diameter, permeability and blood flow, and are able to deconvolve the temporal dynamics of the response using sophisticated imaging techniques. At the center of the study is their newly-developed image analysis tool RVDM (relative velocity, direction and morphology), which allows them to estimate RBC velocities in each vessel. The authors show that the blood velocities agree with vessel classification (artery, capillary, vein), and that intradermal injection of VEGFA caused leakage of dextran with vessel-type dependent kinetics, from capillaries and veins. They conclude that intravital imaging of non-invasive stimulation combined with RVDM analysis allows quantification of rapid events in the vasculature.

While the biological insight provided by the study is limited, there is some novelty in the methods presented. A technique for measuring blood flow on a per-vessel basis with standard confocal or MPM would be very useful for many researchers. However, before this can be considered ready for general use, there are some issues concerning the theory and implementation that should be addressed.

Specifically:

It appears that the authors assume a general shape for all RBCs, computed as an average based on fast scanning microscopy. They use this shape as a “template” to calculate how the shearing that results from cell movement during slow scans distorts the image. This lets them calculate a velocity based on the distortion from the baseline, average shape. It would be useful for the authors to discuss the implications of this assumption in more detail. How much deviation is there in baseline shape, and how much error is introduced by assuming they are all ellipsoidal?

In Figure s1, it would be instructive for the authors to present the binary, segmented version of the vasculature that is used in the analysis.

Some of the images show bright RBCs (e.g., Fig. 2S), but others show dark RBCs (e.g., Fig. 3). Which labeling method is used for RVDM, and why?

In many of the images, the hematocrit appears low, and there is little overlap between RBCs. Does the technique still work in larger vessels with higher hematocrits, where RBCs are packed more closely together?

It is surprising that so much information can be extracted from a single image frame. Are there certain vessel angles that are more accurate than others?

Similarly, are there velocities that are outside of the range of the analysis? It seems that there should be some critical relationship between flow velocity and scan speed that allows or precludes determination.

Finally, I would suggest revising the title. Currently, it is too vague and does not describe well the contents of the study

Reviewer #2 (Remarks to the Author):

This is an exciting paper from Honkura et al that makes use of newly developed and highly sophisticated methodology to provide much more precise measurements to phenomena (e.g., measures of blood flow, vascular dilation and macromolecular permeability) that have been studied for many years with less precise light and fluorescent microscopic methods. The conclusions drawn are consistent with earlier observations but provide data with much greater precision with regard to vessel subtypes and issues of timing in response to effects of VEGFA. Demonstration of the molecular heterogeneity of the microvasculature with regard to claudin and isolectin expression is novel. As the authors indicate, the technology employed will have great utility in subsequent studies of tumor blood vessels and those participating in acute and chronic inflammation. Taken together, this is an excellent piece of work, congratulations to the authors!

I have very little to say by way of criticism but would note the following for the authors' consideration:

- Because the technology used is advanced and highly technical, some points may not be clear to the non-expert reader who does not want to delve deeply into the online methodology. So, for example, it might be good to define more clearly such terms as "XT", "sXYT", "fXYT", etc in the body of the text to help the reader better understand the findings presented.
- It is interesting that over the time span studied arteries did not dilate. One might have expected such dilation due to VEGF-induced eNOS/NO production but very likely the time span was insufficient. Certainly tumor-associated arteries enlarge greatly following exposure to VEGF over a longer time frame.
- The authors nicely correlate macromolecular permeability with vessels that lack the junctional protein claudin. However, this correlation does not necessarily imply exclusive inter-

endothelial cell passage of tracers. Electron microscopic studies from many authors have clearly shown macromolecular tracers passing across microvascular endothelial cells by a transcellular route through vesicles (e.g., very recent papers such as Chow/Gu Neuron 93:1325, 2017 and older works such as the authors' reference #8 and papers cited in that reference). The possibility of trans-endothelial cell tracer passage should at least be mentioned.

Reviewer #3 (Remarks to the Author):

Summary

A method is proposed using slow scan imaging of tissue vasculature to measure both velocity and direction of blood flow of many vessels in view in a real-time manner by taking into account the distortion of the red blood cell images with respect to scan velocity and is demonstrated in the ear dermis of mice. The method is verified using both line scan methods and relative velocity field scanning (RVFS) and used to measure changes in vascular dynamics in a real-time approach when vascular leakage is induced by VEGFA164. Findings from VEGFA164 injection indicate that dilation and flow reduction, occurring approximately two minutes post-injection, cannot be uncoupled. Additionally, the paper investigates the expression of the tight junction protein, Claudin 5, across different vessel types, including arterioles, arteriole side capillaries, venule side capillaries, and venules, and links leakage behaviors after VEGFA administration to Claudin 5 expression levels.

Major Comments and Recommendation

The primary novelty presented in this paper is the method for imaging vascular flow across many vessels at once in two-photon image acquired using relatively standard acquisition parameters. This technique is then applied as one of several different kinds of measurements to explore the details of the vascular response to VEGFA administration. The findings, that vessels leak, dilate, and then increase flow speed in response to VEGFA is not surprising, although the characterization of the kinetics here are well done. The work identifying leakage points in vessels that are primarily present in vessels not expressing the tight junction protein Claudin 5 is also a nice characterization that is not unanticipated.

Strikingly, the authors do not make use of the RVDM analysis to simply characterize blood flows across a vascular network. This seems like an obvious application and demonstration of the power of the approach. Adding such work would enable the authors to frame the paper as being more about the flow measurement method, with a couple of applications – mapping flows across a broad network and (the part that is already completed) simultaneously measuring multiple parameters (flow, leakage, diameter, etc.) in response to a stimulus because the flow measurement does not require unique scan parameters. At a minimum, the authors could map blood flow across a network of vessels captured in a single planar image. Better yet would be to take image stacks and use the method frame by frame to reconstruct flow speeds throughout a three-dimensional vascular network (either represented with a velocity-based colorscale or vector field). This data would highlight the capabilities of this flow measurement approach much better than the results currently presented. [

Additionally, this reviewer feels it is essential for the authors to include their analysis code, example data, and a simple walkthrough/troubleshooting guide as supplementary materials so other researchers may quickly adopt the methodology.

With these changes, a revised version of the manuscript could be considered for Nature Communications.

Minor comments

- Introduction
 - Should clarify that time resolution is impacted in RVFS due to requiring scans at multiple speeds and angles
- Discussion
 - Limitations from requiring calibration using fXYT are not mentioned in discussion. Inferred from the methods. Would this always be required, or would it potentially differ in different tissue beds?
 - Discuss limitations in disease models that may alter blood cell morphology, e.g. sickle cell
 - Upper limit of detectable velocity implied but not stated
 - o In cortex, diameter ranges from 50um and smaller, blood flow ranges from 50um/s up to 15mm/s
 - o Figure 3C showed measurement reliability up to 200-250um/s tested

- o What happens if velocity is higher than threshold?

- Other comments
 - Last statement in “Blood vessel classification based on kinetics...” can be combined with previous paragraph
 - Topic sentence of section titled “Dynamics of vascular leakage” feels too generalized and not focused on showing results for dynamics investigated
 - In many figures, p-value stated as “ $p > 0.0X$.” Should it be “ $p < 0.0X$?”
 - Phrasing of “equations (3) and (4) were then used calculate mean RBC...” change to “equations (3) and (4) were then used to calculate mean RBC...” in RBC flow velocity calculation section
 - Visual aid for rejection of RBC orientations in RBC flow velocity calculation section

We have gone through the reviewers' comments in detail; please find below a point-to-point response to their comments. We have taken care to react to all comments, which we found constructive and valuable. A heatmap indicating blood flow velocities across a planar or vascular network as requested is provided. We now also submit a Walkthrough and troubleshooting guide together with an excel-format software for RVDM analysis to make the RVDM resource generally available. In the Methods section, we have used the heading "RVDM availability" instead of "Code availability" to describe how to access the software and the guide. If the heading is not appropriate, it can of course be adjusted. Changes in the text are marked in yellow. In addition, the text has been carefully revised for grammar and readability (in particular the Introduction). Such grammatical changes are not marked.

In addition to the amendments in response to the reviewers' concerns, we have added more information about Cldn5, the phenotype of the knockout mouse and aspects concerning Cldn5 distribution, in the discussion.

Reviewer #1 comments:

We thank the reviewer for the careful description of our work and the useful comments.

1. It appears that the authors assume a general shape for all RBCs, computed as an average based on fast scanning microscopy. They use this shape as a "template" to calculate how the shearing that results from cell movement during slow scans distorts the image. This lets them calculate a velocity based on the distortion from the baseline, average shape. It would be useful for the authors to discuss the implications of this assumption in more detail. How much deviation is there in baseline shape, and how much error is introduced by assuming they are all ellipsoidal?

Response:

When imaging RBCs using a fixed objective lens, error will certainly be introduced due to the changing external forces influencing RBC rotation and orientation. The assumption that RBCs have an ellipsoidal shape represents an average shape and comes from numerous observations in vessels of various velocities using rapid acquisition. Thus, we believe that this represents an accurate baseline shape to which we can compare distorted images acquired by slow acquisition. The exact shape and motion of RBCs under flow is still a matter of research but this ellipsoidal shape is in keeping with published work examining erythrocyte dynamics when subject to shear flow (for example see Dupire *et al.*, *Full dynamics of a red blood cell in shear flow. Proc Natl Acad Sci U S A. 2012 Dec 18;109(51):20808-13*). The error that is introduced in the baseline dimensions from measurements carried out on fXYT images is now shown in the new Figure S2c, demonstrating that most error is seen in the axis orthogonal to flow (2b) due to the rotations of RBCs around the parallel axis. Furthermore, these points are now better described in the Discussion, pages 10-11 and Methods, pages 16-18.

2. In Figure s1, it would be instructive for the authors to present the binary, segmented version of the vasculature that is used in the analysis.

Response:

Figure S1 shows the distribution of different vessel types and their proportion when segregated by diameter. To measure this diameter FWHM analysis was used which is not compatible with binary images and thus the image shown in Figure S1 does represent one used in this analysis.

3. Some of the images show bright RBCs (e.g., Fig. 2S), but others show dark RBCs (e.g., Fig. 3). Which labeling method is used for RVDM, and why?

Response:

RBCs were not labeled in this study; instead, RBC images were visualized directly. The RBC image rendering in 2-photon results in a dark shape compared to the fluorescent dextran within the plasma as shown in Fig. 3. In Fig. S2a, we inverted the image to better illustrate the RBC shape under different flow parameters. This is now clarified in the figure legend which reads “Typical inverted image of an RBC under very slow blood flow.”

4. In many of the images, the hematocrit appears low, and there is little overlap between RBCs. Does the technique still work in larger vessels with higher hematocrits, where RBCs are packed more closely together?

Response:

We recognize that higher hematocrit counts can make analysis difficult. This will be more apparent in large arteries where erythrocyte distortion is largest. In veins, the slower flow speed means that distinguishing one erythrocyte from another is not difficult. However, even in large arteries measurements can be carried out accurately given careful imaging and analysis (see the new supplemental file “Walkthrough and troubleshooting”). Firstly, in this study the use of two-photon microscopy gives focal resolution of the vertical axis of about 1.2 μm , thus minimizing the occurrence of vertical erythrocyte overlap in images. Secondly, analysis was only carried out on those erythrocyte images that were clear and distinct, thus again minimizing the occurrence and influence of adjacent erythrocytes (see Methods, page 16 and “Walkthrough and troubleshooting guide”). Lastly, the rejection of measurements that differ by more than $\pm 34.1\%$ between V_x and V_y after angle compensation and (or) that fall outside 2σ ($\theta=15-75^\circ$) or 1σ ($\theta=0-15^\circ$ and $75-90^\circ$) within the distribution of repeated measurements further reduces the impact of adjacent erythrocytes (see final two sentences under heading “RVDM flow velocity calculations” in Methods, page 18). These considerations are important for the analysis and have therefore also been mentioned in the Discussion, page 11.

5. It is surprising that so much information can be extracted from a single image frame. Are there certain vessel angles that are more accurate than others? Similarly, are there velocities that are outside of the range of the analysis? It seems that there should be some critical relationship between flow velocity and scan speed that allows or precludes determination.

Response:

Indeed, certain vessel angles do allow more accurate measurements than others. Flow directions against the direction of laser scanning produce less accurate numbers due to the shrinkage, rather than enlargement, of erythrocyte image size (thus reducing the pixel numbers toward the optical limit). Furthermore, analysis accuracy is impacted by the value of θ , with those possessing values of 0, 90, 180 and 270° giving lower accuracy because erythrocyte distortion can only be measured in one axis. These points have now been described in the Discussion, page 11, first paragraph, and the Walkthrough and troubleshooting guide.

In addition, there are velocities that can fall outside of the range of the analysis, but this is dependent on the acquisition parameters. Slower flow velocities can result in no measureable erythrocyte distortion. However, these can be analyzed simply by measuring their distanced travelled between frames. On the other hand, flow velocities that are faster than the laser scan

speed can also result in low accuracy analysis dependent on flow angle and rate of scanning. These points have now been described in the Discussion, first paragraph, page 11, and the Walkthrough and troubleshooting guide.

6. Finally, I would suggest revising the title. Currently, it is too vague and does not describe well the contents of the study

Response:

We agree and have changed the title to: “**Novel methods for vessel identification and assessment of concurrent dynamic vascular events**”.

Reviewer #2 comments:

We thank the reviewer for his/her appreciation of our work and the remark that “The conclusions drawn are consistent with earlier observations but provide data with much greater precision with regard to vessel subtypes and issues of timing in response to effects of VEGFA. Demonstration of the molecular heterogeneity of the microvasculature with regard to claudin and isolectin expression is novel. As the authors indicate, the technology employed will have great utility in subsequent studies of tumor blood vessels and those participating in acute and chronic inflammation. Taken together, this is an excellent piece of work, congratulations to the authors!”

1. Because the technology used is advanced and highly technical, some points may not be clear to the non-expert reader who does not want to delve deeply into the online methodology. So, for example, it might be good to define more clearly such terms as “XT”, “sXYT”, “fXYT”, etc in the body of the text to help the reader better understand the findings presented.

Response:

We agree and the different terms have been explained when they appear in the Results section. In addition, we now provide a new supplemental file “Walkthrough and troubleshooting guide”, which will help the non-expert reader to apply the method.

2. It is interesting that over the time span studied arteries did not dilate. One might have expected such dilation due to VEGF-induced eNOS/NO production but very likely the time span was insufficient. Certainly tumor-associated arteries enlarge greatly following exposure to VEGF over a longer time frame.

Response:

We certainly agree with the reviewer that larger arteries should dilate in response to NO production. In the local vascular bed, however, this does not seem to occur. We are very interested in the tumor vasculature and have performed a pilot with tumor cells injected in the ear dermis. Our initial observations on very small tumors show, interestingly, that there is a very clear demarcation around the tumor, outside which the vasculature appears completely normal with regard to morphology and VEGFA-induced leakage. The range of action of tumor-produced VEGF is therefore very restricted. Moreover, we have not been able to find a transition zone where vessels are gradually transformed into the typical hyperpermeable, dilated tumor vasculature. Even on very small tumors, there is no effect of exogenous VEGF since the tumor vessels are already leaking the tail-vein injected large mw dextran. We have decided to continue to experimentally address the features of the tumor vessels for example in response to anti-VEGF and anti-VEGFR2 therapies, however, we consider this to be a separate study. We thank the reviewer for the useful comment.

3. The authors nicely correlate macromolecular permeability with vessels that lack the junctional

protein claudin. However, this correlation does not necessarily imply exclusive inter-endothelial cell passage of tracers. Electron microscopic studies from many authors have clearly shown macromolecular tracers passing across microvascular endothelial cells by a transcellular route through vesicles (e.g., very recent papers such as Chow/Gu Neuron 93:1325, 2017 and older works such as the authors' reference #8 and papers cited in that reference). The possibility of trans-endothelial cell tracer passage should at least be mentioned.

Response:

We agree and apologize for having omitted this important information. We have amended the introduction with more details on in particular earlier work (see Introduction, 3rd paragraph, page 3) and the vesicular route for transvessel passage in the Discussion, 1st paragraph, page 12).

Reviewer #3 comments:

We thank the reviewer for the careful description of our work and useful comments.

1. Strikingly, the authors do not make use of the RVDM analysis to simply characterize blood flows across a vascular network. This seems like an obvious application and demonstration of the power of the approach. Adding such work would enable the authors to frame the paper as being more about the flow measurement method, with a couple of applications – mapping flows across a broad network and (the part that is already completed) simultaneously measuring multiple parameters (flow, leakage, diameter, etc.) in response to a stimulus because the flow measurement does not require unique scan parameters. At a minimum, the authors could map blood flow across a network of vessels captured in a single planar image. Better yet would be to take image stacks and use the method frame by frame to reconstruct flow speeds throughout a three-dimensional vascular network (either represented with a velocity-based colorscale or vector field). This data would highlight the capabilities of this flow measurement approach much better than the results currently presented.

Response:

We thank the reviewer for this advice and now present a color-coded 3D velocity map in the new Fig S3, This map has been constructed from the broad vascular network shown in Fig S1, which was acquired using XYZ acquisition with tile scan to capture a wide field of view. This approach illustrates the simple imaging parameters required for RVDM analysis.

2. Additionally, this reviewer feels it is essential for the authors to include their analysis code, example data, and a simple walkthrough/troubleshooting guide as supplementary materials so other researchers may quickly adopt the methodology.

Response:

The analysis software in excel format and a Walkthrough and troubleshooting guide are now presented as a supplement; it was indeed our intention to make the method generally available. We thank the reviewer for this suggestion.

Minor comments

3. Introduction

Should clarify that time resolution is impacted in RVFS due to requiring scans at multiple speeds and angles

Response:

The following sentence has been added in the Introduction, page 4.

“As it requires scans at multiple angles, RVFS is not however compatible with simultaneous visualization of other concurrent rapid processes.”

4. Discussion

Limitations from requiring calibration using fXYT are not mentioned in discussion. Inferred from the methods. Would this always be required, or would it potentially differ in different tissue beds?

Response:

The RBC dimensions presented in this paper will be applicable in the majority of vascular beds within the same strain. We cannot rule out however that the high shear experienced by RBC's within very fast flowing vessels such as aorta, or that species or strain variation will result in a change in these dimensions. This has been addressed in the Discussion, 3rd paragraph, page 11, and instructions for how to calculate these dimensions using fXYT are given in the Walkthrough and troubleshooting guide.

5. Discuss limitations in disease models that may alter blood cell morphology, e.g. sickle cell

Response:

Such situations of altered RBC morphology may be overcome by re-calibration of RBC dimensions using fXYT, the instructions for which are now in the supplemental Walkthrough and troubleshooting guide. However, the behavior of such RBCs under flow may differ greatly from that of normal RBCs and so introduce greater error. These points have been added in the Discussion, 3rd paragraph, page 10, and instructions for re-calibration given in the Walkthrough and troubleshooting guide.

6. Upper limit of detectable velocity implied but not stated. In cortex, diameter ranges from 50um and smaller, blood flow ranges from 50um/s up to 15mm/s

Response:

The RVDM analysis in itself has no limit of detectable velocity. Limitations however lie in the scan speed with which imaging can be carried out and the achievable resolution and signal-to-noise ratio. RVDM could be used to measure flow speeds much faster than those presented here but would however require different imaging parameters such as restriction of field size. This advice has been added to the Discussion, page 1 and the Walkthrough and troubleshooting guide.

7. Figure 3C showed measurement reliability up to 200-250um/s tested

Response:

For the tissue studied here 250 $\mu\text{m/s}$ was found to be the upper range of vessel velocity. Using our imaging parameters to allow imaging of a wide field of view, 300-400 $\mu\text{m/s}$ is close to the upper limit of what can be analyzed whilst still including all flow directions. As stated above, faster acquisition however will allow analysis of faster flow speeds. This is now addressed in the Discussion, 1st paragraph, page 11.

8. What happens if velocity is higher than threshold?

Response:

When flow velocities exceed the threshold of acquisition parameters, erythrocytes will be represented as a shape with a length under the optical limit that cannot be measured accurately. This however can be overcome by using faster scanning speeds.

Furthermore, it is possible in the rare situation when the flow speed is the same as the scan speed and $\theta = 90^\circ$ or 270° that RBCs don't appear in the image. Again, this can be overcome by using faster scanning speeds. We now address these aspects in the Discussion, 1st paragraph, page 11, and in the Walkthrough and troubleshooting guide.

9. Other comments

- Last statement in “Blood vessel classification based on kinetics...” can be combined with previous paragraph.

Response: Amended.

- Topic sentence of section titled “Dynamics of vascular leakage” feels too generalized and not focused on showing results for dynamics investigated

Response: Amended.

- In many figures, p-value stated as “ $p > 0.0X$.” Should it be “ $p < 0.0X$?”

Response: We apologize for this error, which has been amended.

- Phrasing of “equations (3) and (4) were then used calculate mean RBC...” change to “equations (3) and (4) were then used to calculate mean RBC...” in RBC flow velocity calculation section:

Response: Amended.

10. Visual aid for rejection of RBC orientations in RBC flow velocity calculation section

Response:

There aren't any specific RBC orientations that are rejected as RVDM is able to calculate the velocity of RBCs flowing in any orientation. RBC velocity values are rather rejected if the velocities calculated from the X and Y distortions (V_x and V_y) differ by more than 34.1% after angle compensation and (or) they fall outside 2σ ($\theta = 15-75^\circ$) or 1σ ($\theta = 0-15^\circ$ and $75-90^\circ$) within the distribution of repeated measurements. This information has been added in Methods, 1st paragraph, page 18. The reader will be further guided by the new supplemental Walkthrough and troubleshooting guide.

With these corrections, new text additions, as well as two novel figures (Suppl 1 and 3) together with the new Walkthrough and troubleshooting guide and the RVDM software, we hope to have responded to the reviewers' comments in a satisfactory manner. We thank the editor and reviewers for their constructive comments that helped to improve our work.

We look forward to hearing from you about your decision.

Sincerely yours,
Lena Claesson-Welsh

REVIEWERS' COMMENTS:

Reviewer #1 (Remarks to the Author):

Reviewer 1: In Figure s1, it would be instructive for the authors to present the binary, segmented version of the vasculature that is used in the analysis.

Author Response: Figure S1 shows the distribution of different vessel types and their proportion when segregated by diameter. To measure this diameter FWHM analysis was used which is not compatible with binary images and thus the image shown in Figure S1 does represent one used in this analysis.

Reviewer 1 reply: The text should more clearly state that the analyses are being performed manually rather than via automated computer algorithms. This raises additional concerns, such as how the operator determines how to draw the perpendicular line to measure FWHM. Is this step operator dependent? What is the error / variability?

Reviewer 1: It is surprising that so much information can be extracted from a single image frame. Are there certain vessel angles that are more accurate than others? Similarly, are there velocities that are outside of the range of the analysis? It seems that there should be some critical relationship between flow velocity and scan speed that allows or precludes determination.

Author Response: Indeed, certain vessel angles do allow more accurate measurements than others. Flow directions against the direction of laser scanning produce less accurate numbers due to the shrinkage, rather than enlargement, of erythrocyte image size (thus reducing the pixel numbers toward the optical limit). Furthermore, analysis accuracy is impacted by the value of θ , with those possessing values of 0, 90, 180 and 270° giving lower accuracy because erythrocyte distortion can only be measured in one axis. These points have now been described in the Discussion, page 11, first paragraph, and the Walkthrough and troubleshooting guide. In addition, there are velocities that can fall outside of the range of the analysis, but this is dependent on the acquisition parameters. Slower flow velocities can result in no measureable erythrocyte distortion. However, these can be analyzed simply by measuring their distanced travelled between frames. On the other hand, flow velocities that are faster than the laser scan speed can also result in low accuracy analysis dependent on flow angle and rate of scanning. These points have now been described in the Discussion, first paragraph, page 11, and the Walkthrough and troubleshooting guide.

Reviewer Reply: Because the novelty and power of the study lies in the accuracy and accessibility of the velocity measurements, these points need further clarification.

1) Given that each RBC is measured manually, how does the operator choose which ones to measure, and how operator-dependent is this? How long does it take to measure a field of vessels?

2) In a given field of vessels (e.g., figure S3), how many vessels can be measured via the “in-frame” method, and how many have to be tracked directly (“frame” method)?

3) In a given field of vessels (e.g., figure S3), how much uncertainty is there in the velocity for each vessel? And how many can't be determined reliably? The authors write that the code returns standard deviations, but there is no clear depiction of how these are affected by vessel angle or flow direction.

4) How rapidly could additional scan angles or speeds be taken, and how would this increase the accuracy of method?

Finally, the authors should consider again the advantages and disadvantages of their approach compared with other, more automated and unbiased methods. There are major drawbacks to their technique that are glossed-over in the manuscript.

Reviewer #2 (Remarks to the Author):

no further comments

Reviewer #3 (Remarks to the Author):

The authors have adequately addressed my previous concerns and I encourage publication in Nature Communications.

Reviewer #1 (Remarks to the Author):

1. The text should more clearly state that the analyses are being performed manually rather than via automated computer algorithms. This raises additional concerns, such as how the operator determines how to draw the perpendicular line to measure FWHM. Is this step operator dependent? What is the error / variability?

Response. While the image analysis itself largely relies on computer algorithms, the imaging of the tissue by necessity involves selection of the area to be imaged and the cells/structures to be analyzed and used as a basis for the image analysis. In order for the imaging and the subsequent analysis to be entirely automated, the tissue or structure would have to be extremely defined and fixed. In contrast, image analysis methods for complex dynamic events such as blood flow velocity are incompatible with automatic image algorithms. Please consult the available applications for measuring blood flow velocity on fluorescent images such as the residence time line scanning (RLTS) and relative velocity field scanning (RVFS). Compared to these methods, the RVDM has several added advantageous features as described in the paper. Please see first paragraph in the Discussion, page 9.

The reviewer specifically asks how the operator draws the perpendicular line to measure FWHM. The FWHM method is very well known and applied widely across fields to measure the length of very small objects independently of background signals. For a reference on FWHM, please see chapter 1 in "Introduction to Confocal Fluorescence Microscopy" and ref 38 in the reference list.

Following the well-established procedure for FWHM, the perpendicular line is drawn 90° to the direction of blood flow. This offers a more accurate alternative to drawing a line perpendicular to the vessel edge as it is unaffected by the signal intensity, noise and the changes in the observed fluorescence following vessel leakage. The perpendicular line should be sufficiently longer than the vessel (approximately 2x longer) to provide the algorithm with an accurate baseline. Furthermore, to allow accurate classification of vessel identity, multiple measurements should be taken along a vessels length and the average taken. We realize that this might be important information for future, potentially inexperienced users. We have therefore amended the materials and methods to give more detailed instructions for vessel diameter measurement. Regarding error, the FWHM measurement has no error and was only used in this study as an aid to classify vessels as capillaries or venules/arterioles along with the dye bolus injection and RBC velocity. The distribution of blood vessel diameters within a vascular field as shown in Supplementary Figure 1b is expressed as SD.

2. Because the novelty and power of the study lies in the accuracy and accessibility of the velocity measurements, these points need further clarification.

Given that each RBC is measured manually, how does the operator choose which ones to measure, and how operator-dependent is this? How long does it take to measure a field of vessels?

Response: The RBC flow angle measurements that form the basis for the RVDM flow estimations were compared to those acquired using line (XT) and fast scan (fXYT) methods. As is shown in Fig. 3C, the estimates agree between these methods, even at flow angles close to 0, 90, 180 and 270°. Importantly, only the RVDM allows estimations before, during and after stimulation.

Because the code to allow RVDM analysis contains stringent rejection criteria, accurate analysis with RVDM is operator independent. It is not required that the operator is selective of which RBC images are measured as those that are, for example, formed of two RBCs or have poor resolution, will be rejected from the final results. It is recommended in the walk-through and trouble-shooting guide that RBC images with a high S/N ratio are used, however this to save the operator's time rather than ensure accurate results.

The time taken to measure a field of vessels is of course dependent on the size of the field, however analysis of the flow velocity of a single vessel will take approximately 2-4 minutes for an experienced operator.

3. In a given field of vessels (e.g., figure S3), how many vessels can be measured via the "in-frame" method, and how many have to be tracked directly ("frame" method)?

Response: This is of course dependent on the scan speed. However, for Figure S3 of the 248 vessels (segments between intersections and bifurcations), 206 required analysis in-frame whilst the remainder were compatible with frame analysis.

4. In a given field of vessels (e.g., figure S3), how much uncertainty is there in the velocity for each vessel? And how many can't be determined reliably? The authors write that the code returns standard deviations, but there is no clear depiction of how these are affected by vessel angle or flow direction.

Response: An example of the error of the flow velocity of individual vessels can be seen in Figure 3c. This is within the range of 5-15% of the velocity, similar to the error of the much more established techniques XT and fXYT. Vessel velocity can be measured reliably using RVDM as long as the flow velocity is compatible with the scan speed and that RBCs are flowing through. In Figure 3c, which was taken as a single Z-stack, 1% of capillaries had no RBCs flowing through and thus could not be measured. The acquisition of this field as a time-lapse acquisition, as has been done for all other experiments within this study, would solve this.

Regarding the return of standard deviations, whilst as stated, the analysis of flow angles near $\theta = 0, 90, 180$ and 270° is more difficult, error for values of $\theta = 0-15^\circ, 75-105^\circ, 165-195^\circ, 255-285^\circ$, and $345-360^\circ$ is minimized as these flow angles are subject to a more stringent selection within the distribution of measurements. The flow angles of the individual vessels are provided for the reviewer's benefit in the figure below (Fig. 3c in main paper with flow angles indicated). It should be noted that whilst these vessel angles are more difficult, single vessels within a field are often not straight and thus other sections of a

vessel segment may be chosen for measurement that have a more favorable flow angle. This is be stated in the walk-through and trouble-shooting guide.

5. How rapidly could additional scan angles or speeds be taken, and how would this increase the accuracy of method?

Response: As described above, additional scan angles would not be necessary to carry out accurate RVDM quantification. On the other hand, additional scan speeds could be useful for vascular beds that have a very broad range of flow speeds. In this case we believe that only two scan speeds would be necessary, one at a normal scan speed as we have used in this study and one faster to allow analysis of very fast flowing arteries. The possible need for multiple scan speeds has no influence on the accuracy of the method however. If the velocity is within the range where whole RBC images are visible at a given scan speed, then the method allows accurate measurement.

6. Finally, the authors should consider again the advantages and disadvantages of their approach compared with other, more automated and unbiased methods. There are major drawbacks to their technique that are glossed-over in the manuscript.

Response: With the addition of selection criteria for the rejection of those measurements that fall outside strict parameters, RVDM is highly unbiased and unaffected by user error. We recognize that the disadvantage of RVDM is the need for user time. However, RVDM offers many advantages not allowed by other current techniques, the primary of which is its compatibility with time lapse acquisition of large fields of view, thus allowing the capture of concurrent, dynamic processes and the correlation of blood flow velocity with other vascular events. We have added a short comment in the discussion concerning the degree of complexity of the processes analyzed in relation to automatization, see first paragraph, page 11.

We hope that our explanations have adequately taken care of the concerns of Reviewer 1.

Sincerely,

Lena Claesson-Welsh

Comparison of RBC velocities estimated from sXYT combined with RVDM, XT and fXYT image acquisition in the same vessels. Flow angles are indicated for each measurement. $n=3$ mice with more than 20 measurements/vessel.